# Bacterial Volatiles (mVOC) Emitted by the Phytopathogen *Erwinia amylovora* Promote *Arabidopsis thaliana* Growth and Oxidative Stress

**DOI:** 10.3390/antiox12030600

**Published:** 2023-02-28

**Authors:** Ambra S. Parmagnani, Chidananda Nagamangala Kanchiswamy, Ivan A. Paponov, Simone Bossi, Mickael Malnoy, Massimo E. Maffei

**Affiliations:** 1Department of Life Sciences and Systems Biology, University of Turin, Via Quarello 15/a, 10135 Turin, Italy; 2Research and Innovation Centre, Edmund Mach Foundation, Via Edmund Mach 1, 38098 San Michele all’Adige, Italy; 3Department of Food Science, Aarhus University, 8200 Aarhus, Denmark

**Keywords:** microbial volatile organic compounds, 1-nonanol, 1-dodecanol, calcium signaling, potassium channels, ROS and NO burst, transcriptomics, PIN1 and PIN3 efflux carriers, abscisic acid, auxin

## Abstract

Phytopathogens are well known for their devastating activity that causes worldwide significant crop losses. However, their exploitation for crop welfare is relatively unknown. Here, we show that the microbial volatile organic compound (mVOC) profile of the bacterial phytopathogen, *Erwinia amylovora*, enhances *Arabidopsis thaliana* shoot and root growth. GC-MS head-space analyses revealed the presence of typical microbial volatiles, including 1-nonanol and 1-dodecanol. *E. amylovora* mVOCs triggered early signaling events including plasma transmembrane potential Vm depolarization, cytosolic Ca^2+^ fluctuation, K^+^-gated channel activity, and reactive oxygen species (ROS) and nitric oxide (NO) burst from few minutes to 16 h upon exposure. These early events were followed by the modulation of the expression of genes involved in plant growth and defense responses and responsive to phytohormones, including abscisic acid, gibberellin, and auxin (including the efflux carriers PIN1 and PIN3). When tested, synthetic 1-nonanol and 1-dodecanol induced root growth and modulated genes coding for ROS. Our results show that *E. amylovora* mVOCs affect *A. thaliana* growth through a cascade of early and late signaling events that involve phytohormones and ROS.

## 1. Introduction

The economic importance of the Gram-negative bacterium *Erwinia amylovora* (fire blight, Enterobacteriaceae) is increasing with the outbreak of old disease bacterial speck of apple and pear and other commercial and ornamental Rosaceae host plants [1]. *E. amylovora* secretes effectors and proteins that are involved in the hypersensitive response and pathogenicity [2]. Plants are sessile, multicellular organisms that rely on developmental and metabolic changes for growth. Extensive communication occurs between plants and microorganisms during different stages of plant development, in which signaling molecules from the two partners play an important role [3,4]. Fungal and bacterial species are able to detect the plant host and initiate their colonization strategies in the rhizosphere by producing canonical plant growth-regulating substances such as auxins or cytokinins [5]. On the other hand, plants are able to recognize microbe-derived compounds and adjust their defense and growth responses according to the type of microorganism encountered [6]. This molecular dialogue will determine the final outcome of the relationship, ranging from pathogenesis to symbiosis, usually through highly coordinated cellular processes [7]. Bacterial and fungal phytopathogens are not restricted to infecting shoot or root tissues exclusively, and as such, communication between the shoot and root can confer a survival advantage to the plant and potentially limit or prevent diseases. For instance, beneficial bacteria and fungi can confer immunity against a wide range of foliar diseases by activating plant defenses, thereby reducing a plant’s susceptibility to pathogen attack [3,8]. For many years this phenomenon was considered the basis by which beneficial microorganisms could increase plant growth when inoculated in crops; however, it is quite evident that novel microbial signals play important roles in plant morphogenesis [3].

Plants activate various barriers and defensive strategies to protect against the attack of pathogens [9]. The first line of defense is triggered by the recognition of invariant microbial epitopes known as pathogen-associated molecular patterns (PAMPs) for the recognition of potential pathogens in the innate immune system of both plants and animals that are so-called elicitors [10]. Both animals and plants can recognize invariant PAMPs [11] that are characteristic of pathogenic microorganisms by pattern recognition receptors (PRRs) in the plasma membrane [9]. In leaves and roots, PAMPs recognition triggers nitric oxide (NO) and reactive oxygen species (ROS) production, as well as a complex cascade of MAP kinases that lead to the activation of transcription factors and defense response genes [12].

Recent discoveries reveal that microbial volatile organic compounds (mVOCs) influence plant growth and defense modulation [13,14,15,16], which has been confirmed by several studies using different bacterial strains and plant models; however, the signaling molecules responsible for the observed effects, as well as the molecular mechanisms underneath this phenomenon are poorly understood [17,18,19]. There have been relatively few studies to detect and understand how plants recognize the presence of bacterial volatiles, how they prime themselves to "get ready" for phytopathogens, and how they promote symbiont association. Therefore, it is important to understand these mechanisms in order to develop sustainable strategies to combat pathogen attacks on food/feed crop plants. 

In this study, we aim to understand the mechanisms underpinning *Arabidopsis thaliana*’s early and late responses to *Erwinia amylovora* volatiles. Thereby, we characterized *E. amylovora* mVOCs and investigated the effects of bacterial volatiles on plasma membrane depolarization, cytosol fluctuations of Ca^2+^ concentration, K^+^ efflux, and levels of ROS and NO in shoots and roots of *A. thaliana* plants. Furthermore, we carried out gene expression analysis, identifying the most differentially expressed genes (DEGs) regulated by *E. amylovora* volatiles. Finally, we used two synthetic mVOCs to assess their effect on *A. thaliana* growth and ROS gene expression.

## 2. Materials and Methods

### 2.1. Plant and Bacterial Co-Cultivation

*Arabidopsis thaliana* Col0 seeds were surface sterilized (2 min 70% ethanol, 5 min 5% calcium hypochlorite, rinsed four times with autoclaved water) and placed on bipartite squared Petri dish containing in the upper part half-strength Murashige–Skoog medium (MS medium) and in the lower part a nutrient agar II (NA II; peptone from casein 3.5 gL^−1^, peptone from meat 2.5 gL^−1^, peptone from gelatin 2.5 gL^−1^, yeast extract 1.5 gL^−1^, NaCl 5 gL^−1^, and agar–agar 15 gL^−1^, pH 7.2). The seeds were vernalized for 3 days at 4 °C in the absence of light, then placed in a growth chamber (16 h light/8 h darkness/light with a fluorescent light of about 70 μmol photons m^−2^ s^−1^). *Erwinia amylovora* strain E273 was obtained from the bacterial collection of Dr. Malnoy Lab and was added to the co-culture at given times (from 30 min to 16 h). In order to standardize the experiments, the same number of seeds and the same amount of *E. amylovora* CFU were always used. Samples of roots and shoots of *A. thaliana* seedlings were collected at different times of *E. amylovora* mVOCs exposure and immediately frozen in liquid nitrogen and stored at −80 °C.

### 2.2. Head Space Analysis of E. amylovora mVOCs

Extraction of *E. amylovora* (strain E273) mVOC was performed by stir bar sorptive extraction (SBSE) by following the same protocol used to extract truffle volatiles in a previous study [20]. Briefly, a stir bar (SB) was attached to the cover lid of the Petri dish by using an external magnet. After 16 h exposure, the SB was desorbed with a Gerstel (GmbH & Co.KG, Mülheim an der Ruhr, Germany) twister desorption unit (TDU), which was connected to a cooled injection system (CIS) that cryoinjected the desorpted molecules into a gas-chromatograph mass spectrometry system as previously described [20]. mVOCs identification was performed by comparison with spectra in mass spectra databases (NIST^®^), comparison of Kovàts retention indices to the literature data and in-house database, and in some cases, direct GC-MS comparison with authentic standards.

### 2.3. Membrane Potential Determination

*A. thaliana* root and leaf transmembrane potential variations (Vm) were obtained as previously reported [21]. Vm variations were recorded both on a pen recorder and through a digital port of a PC using a data logger. The results of Vm are shown as the average number of at least 50 Vm measurements.

### 2.4. Evaluation of Intracellular Calcium Variations by Confocal Laser Scanning Microscopy Usign Calcium Orange

Confocal laser scanning microscopy (CLSM) was used to localize the intracellular calcium efflux by using the calcium orange dye (stock solution in DMSO, Molecular Probes, Leiden, The Netherlands). Roots and shoots of *A. thaliana* exposed to *E. amylovora* mVOC were sampled from 30 min to 3 h of exposure and, after incubation with 5 µM calcium orange, were observed directly on a Nikon Eclipse C1 spectral CLSM. The microscope operates with a Krypton/Argon laser at 488 nm with a BP of 500–540 nm and an LP of 650 nm. Digital images were analyzed using the NIH image software as described earlier [22]. Controls were represented by the application of 5 µM calcium orange solution to unexposed tissues.

### 2.5. Localization of Voltage-Gated K^+^ Channels (VGKC) and Ligand-Gated or Resting Inward Rectifier K^+^ Channels (LG/RIRKC) Using FluxOR^TM^

VGKC and LG/RIRKC were assayed from 30 min to 3 h by using the FluxOR^TM^ Potassium Ion Channel Kit from Invitrogen (Molecular Probes, Leiden, The Netherlands). mVOC-exposed and unexposed *A. thaliana* seedlings were incubated in the dark for 1 h with 100 µL of loading buffer (deionized water, FluxOR^TM^ assay buffer, and probenecid) by following the manufacturer’s instructions. Just before observation, 50 µL of stimulus buffer (deionized water, FluxOR^TM^ chloride-free buffer, K_2_SO_4_, and Tl_2_SO_4_) was added by following the manufacturer’s instructions. CLSM fluorescence was assayed by Nikon Eclipse C1 spectral CLSM, as previously described [23].

### 2.6. Localization of H_2_O_2_ by CLSM Using Amplex Red

*A. thaliana* seedlings exposed to *E. amylovora* mVOCs and unexposed control seedlings were assayed from 3 h to 16 h after incubation with the dye 10-acetyl-3,7-dihydroxyphenoxazine (Amplex Red) as described earlier [21,24]. Tissues were observed with a Nikon Eclipse C1 spectral CLSM, and scans were recorded using a Laser Ar (458 nm/5 mW; 476 nm/5 mW; 488 nm/20 mW; 514 nm/20 mW), a Laser HeNe 543 nm/1.20 mW, and a Laser HeNe 633 nm/10 mW.

### 2.7. Determination of Nitric Oxide with 4,5-diaminofluorescein Diacetate (DAF-FM DA) Using CLSM

mVOC-exposed and unexposed *A. thaliana* seedlings were assayed from 3 h to 16 h to analyze NO accumulation by CLSM using 50 μL of loading buffer (10 μM DAF-FM DA (Molecular Probes)). The Nikon Eclipse C1 spectral CLSM operated with an argon laser with an excitation wavelength of 488 nm. Emissions were recorded using a 508 to 525 nm band pass filter. Carboxy-2-phenyl-4,4,5,5-tetra-methylimidazolinone-3-oxide-1-oxyl (cPTIO) (Sigma, Milan, Italy), an NO scavenger, was dissolved in DMSO and used at a final concentration of 1 mM. The treated tissues were perfused with cPTIO as described above before being stained with DAF-FM DA.

### 2.8. RNA Extraction from Arabidopsis Shoots and Roots upon Exposure to E. amylovora mVOCs

One hundred milligrams of frozen *A. thaliana* roots and shoots exposed for 16 h to *E. amylovora* mVOCs was ground in liquid nitrogen, and total RNA was isolated and its quality and quantity checked as previously described [25]. Quantification of RNA was also confirmed spectrophotometrically by using a NanoDrop ND-1000 (Thermo Fisher Scientific, Waltham, MA, USA).

### 2.9. cDNA Synthesis and Gene Microarray Analyses (Including MIAME)

Five hundred nanograms of total RNA of four biological replicates was reverse-transcribed into double-stranded cDNA, which was then transcribed into antisense cRNA and labeled with either Cy3-CTP or Cy5-CTP fluorescent dyes for 2 h at 40 °C as reported before [25]. Cyanine-labeled cRNAs were purified using the RNeasy Minikit (Qiagen, Hilden, Germany). Purity and dye incorporation were assessed as previously described [25]. Then, 825 ng of unexposed control Cy3-RNAs and 825 ng of mVOC exposed treated Cy5-RNAs were pooled together and hybridized using the Gene Expression Hybridization Kit (Agilent Technologies, Santa Clara, CA, USA) onto 4 × 44 K *A. thaliana* (v3) Oligo Microarray (Agilent Technologies). The microarray experiment followed a direct 2 × 2 factorial two-color design. This resulted in two-color arrays, satisfying the Minimum Information About a Microarray Experiment (MIAME) requirements [26]. Microarrays were scanned as reported before [25]. The microarray data have been deposited in NCBI’s Gene Expression Omnibus [27] (GSM6994948 to GSM6994951) and are accessible through GEO Series accession number GSE223760 (https://www.ncbi.nlm.nih.gov/geo/query/acc.cgi?acc=GSE223760 (accessed on 26 January 2023)).

### 2.10. Validation of Microarray Data by Real-Time PCR

Validation of microarray data was performed by qPCR, which was performed on a QuantStudio 3 Real-Time PCR System (Applied Biosystems, Foster City, CA, USA) using the same cDNA products obtained as previously described (Section 2.9). The amplification was carried out with the primers listed in Appendix A, while the reaction was performed using the Maxima SYBR Green/ROX qPCR Master Mix Kit (Thermo Fisher Scientific, Waltham, MA, USA) in 25 µL of total volume. The thermocycling was performed using a two-step cycling protocol: 50 °C for 2 min, 95 °C for 10 min, 95 °C for 15 s and 60 °C for 60 s. The last two steps were repeated for 40 cycles. All primers were designed using Primer 3 software. Two different reference genes actin1 (*ACT1*, At2g37620) and the elongation factor 1B alpha-subunit 2 (*eEF1Balpha2*, At5g19510) were used to normalize the results of qPCR; the most stable gene was ACT1. All amplification plots were analyzed with the QuantStudio Design and Analysis software (Applied Biosystem, Foster City, CA, USA) to obtain Ct values. Relative RNA levels were calibrated and normalized with the level of ACT1 mRNA.

### 2.11. Expression Analysis of PIN1 and PIN3

Seeds of *A. thaliana* PIN1::GUS, PIN3::GUS, DR5::GUS, PIN1:GFP, and PIN3:GFP were obtained from the Institut für Biologie II, Freiburg, Germany, in 2006 and multiplied in our labs. Histochemical staining for GUS activity was performed using a modified indigogenic method with 5-bromo-4-chloro-3-indoxyl β-D glucuronide (X-Gluc, Sigma) as substrate according to [28]. For all comparisons between mVOC-exposed samples and controls, identical staining conditions were used. GFP was visualized in 5% glycerol without fixation. Microscopy was done with the same microscopy described above.

Time-course gene expression of *PIN1* (*AT1G73590*) and *PIN3* (*AT1G70940*) was assayed from 3 h to 16 h by qRT-PCR as described in Section 2.10.

### 2.12. Exposure of A. thaliana to Synthetic 1-nonanol or 1-dodecanol

Eight-day-old seedlings of *A. thaliana* cultivated as described in Section 2.1 were grown for 72 h in the presence of 20, 50, and 100 µg of either 1-nonanol or 1-dodecanol (pure standards from Fisher Scientific, Rodano (MI), Italy) dissolved in DMSO. Control plates were treated with DMSO. Agar was carved with a sterilized comb, and sterilized filter paper was placed inside the hole (see Appendix A). 1-Nonanol or 1-dodecanol was dissolved in DMSO to obtain a 100 mM solution, from which aliquots were taken and deposited on the filter paper to obtain the desired quantity. Plates were scanned just after the insertion of the volatiles and after 72 h. Then roots and shoots of the seedlings were separately collected and immediately frozen in liquid nitrogen for further analyses. Root and shoot morphology were analyzed with NIH ImageJ software (https://imagej.net/ij/, accessed on 3 January 2023).

### 2.13. Gene Expression of RBOHH, SOD1, CAT1, PER4 and APX1 of A. thaliana Exposed to Synthetic mVOCs

One hundred milligrams of frozen *A. thaliana* roots and shoots exposed for 72 h to different concentrations of 1-nonanol or 1-dodecanol was ground in liquid nitrogen and total RNA was isolated and quantified as described in Section 2.8. Five hundred nanograms of total RNA of three biological replicates was reverse-transcribed into double-stranded cDNA which was transcribed into antisense cRNA and labeled as described in Section 2.9. The amplification was carried out with the primers listed in Appendix A, and the reaction was performed as described in Section 2.9. The reference genes actin1 (*ACT1*, At2g37620) and the elongation factor 1B alpha-subunit 2 (*eEF1Balpha2*, At5g19510) were used as described in Section 2.9, and the relative RNA levels were calibrated and normalized with the level of *ACT1* mRNA.

### 2.14. Statistical Analysis

At least three biological replicates were performed for the CLSM, qRT-PCR, and GC-MS analyses. The mean value was calculated along with the SD. A stem-and-leaf function of Systat 10 was used to treat Vm data to extract the lower and upper hinges from the Gaussian distribution. After filtering the data, the mean value was calculated along with the SD. Processing and statistical analysis of the microarray data were performed in GeneSpring software and with R using the Bioconductor package limma [29]. The raw microarray data are subjected to background subtraction and lowess normalization. Agilent control probes were filtered out. The GeneSpring software was used for finding differentially expressed genes (DEGs). GO enrichment information for the differently expressed probe sets was obtained from The *A. thaliana* Information Resource (https://www.arabidopsis.org/index.jsp (accessed on 3 January 2023)). The protein–protein association and interaction of the DEGs identified in this work were obtained using STRING [30] and by considering known, predicted, and other types of interactions.

## 3. Results and Discussion

### 3.1. Erwnia Amylovora mVOCs Promote Arabidopsis Shoot and Root Growth

To investigate the effect of mVOC on *A. thaliana* growth, we cultivated *A. thaliana* seedlings in the presence of *E. amylovora* mVOCs on the same square Petri dishes with no physical contact between them. This method, hereafter referred to as headspace co-cultivation (HSCC), allows the cultivation of plants and microorganisms in the same environment by using specific culture media. To overcome any effect on plant health and development induced by carbon dioxide accumulation from bacterial culture in HSCC [19], a porous paper tape, which allows circulation of air from outer space and avoids accumulation of mVOCs and CO_2_, was used to seal the Petri dishes.

In response to *E. amylovora* mVOCs and after continuous HSCC for 8 days, *A. thaliana* shoot and root biomass increased with enhanced lateral roots and root hair production (Figure 1), with respect to controls. In particular, a significant difference was found between shoot area expressed as mm^2^ (mean difference =−16.82, 95.00% CI = −18.62 to −15.03, SD difference = 2.33, t = −21.64, *p* < 0.001), primary root length expressed as mm (mean difference = −6.15, 95.00% CI = −7.46 to −4.84, SD difference = 1.95, t = −10.47, *p* < 0.001) and the number of secondary roots (mean difference = −6.89, 95.00% CI = −7.70 to −6.08, SD difference = 1.05, t = −19.61, *p* < 0.001) between controls and *A. thaliana* seedlings exposed to *E. amylovora* mVOCs. This phenology suggests that seedlings perceive mVOCs emitted by *E. amylovora* and the observed morphological response implies the involvement of transcriptional, metabolomic, and physiological modulation. Similar effects have been observed in *A. thaliana* seedlings exposed to mVOCs emitted by plant growth-promoting rhizobacteria from avocado trees [31], after exposure to mVOCs of *Trichoderma asperelloides* PSU-P1 [32] and *Bacillus subtilis* GB03 [33].

### 3.2. E. amylovora Emits a Complex Blend of mVOCs

To analyze the mVOC composition of *E. amylovora,* we cultivated the bacteria in the same Petri dishes without the presence of *A. thaliana* seedlings and used polydimethyl siloxane (PDMS) stir bar sorptive extraction (SBSE) to collect bacterial volatiles.

After 16 h exposure, the chemical composition of *E. amylovora* mVOCs is characterized by the presence of volatile hydrocarbons (pentadecane and heptadecane), which along with ketones (6,10,14-trimethyl-2-pentadecanone and 6,10-dimethyl-2-undecanone) represent the highest percentage of mVOCs emitted (about 26% and 25%, respectively). Alcohols (particularly 1-dodecanol, 1-undecanol, and 1-nonanol) represent 23% of total mVOCs and are followed by aldehydes (above all 1-pentadecanal, tridecanal, tetradecanal, and undecanal), which account for 22% of total mVOCs. Two alkenes (1-tetradecene and 1-hexadecene) and a monoterpene alcohol (1,8-cineole) were also present but at concentrations below 1% of total mVOCs (Table 1). 

The occurrence of pentadecane in mVOC is widespread both in bacteria [34] and in fungi [35] and stimulates plant growth [36], whereas heptadecane emission was found mainly in bacteria [37]. The mVOC 6,10,14-trimethyl-2-pentadecanone (also known as hexahydrofarnesylacetone) is emitted by different microorganisms, including *Pseudomonas syringae* and *Stigmatella aurantiaca* [37,38] and stimulates root production [39]. 1-Dodecanol is emitted by several bacteria [40], whereas 1-undecanol was found in the mVOCs of proteolytic strains of *Serratia proteamaculans* and *Pseudomonas fragi* [41]. Emission of the mVOC 1-nonanol was found to be associated with the *A. thaliana* lateral root formation [31]. Tridecanal and Tetradecanal are also emitted by *Carnobacterium divergens* [41] and by the fungus *Tuber borchii* [42], and induce root formation [39], whereas undecanal is typically emitted by several bacteria [36]. The mVOC 1-tetradecene has been found in *C. divergens* and in *Pseudomonas putida* emissions [41,43], whereas 1-hexadecene was detected in the mVOCs of *C. divergens*, *P. fragi,* and several other bacteria [41]. The monoterpene 1,8-cineole is typically emitted by fungi [44]; however, a bacterial sesquiterpene cyclase similar to a plant homolog from *Streptomyces clavuligerus* was found to act as a monoterpene cyclase, being able to catalyze the conversion of the sesquiterpene precursor geranylgeranyl diphosphate into 1,8-cineole [45].

### 3.3. E. amlylovora mVOCs Induce Early Signaling Events in Arabidopsis thaliana

To evaluate the early responses of *A. thaliana* to *E. amylovora* mVOCs, we set up experiments aimed at evaluating short-time exposures of seedlings to bacterial mVOCs. In both roots and shoots, after evaluating the electrophysiological changes in the membrane potential, we used confocal laser scanning microscopy (CLSM) with the use of specific fluorescent indicators to assess the content of calcium, the activity of different potassium (K^+^) channels, and the presence of reactive oxygen species (ROS) and nitric oxide (NO).

#### 3.3.1. *E. amylovora* mVOCs Depolarize *A. thaliana* Plasma Membrane Potential (Vm)

Early events upon pathogen infection include alteration of the plasma transmembrane potential (Vm), due to unbalanced ion fluxes across the plasma membrane, influx of Ca^2+^ into the cytosol from the cell wall and internal stores, K^+^-gated channel activity, and ROS and NO burst [46]. Time-course experiments were performed in order to assess the effect of *E. amylovora* mVOCs on *A. thaliana* leaf and root Vm. A strong Vm depolarization was found between 30 min and 6 h of mVOCs exposure in *A. thaliana* leaves (Figure 2). A similar trend was observed for *A. thaliana* root Vm depolarization, although with a significantly (*p* < 0.05) reduced response from 30 min to 6 h exposure. Longer mVOC exposure times induced a significant but reduced Vm depolarization in both leaves and roots (Figure 2).

Leaf and root cells have developed specific chemosensorial mechanisms to recognize chemical signals at very low concentrations that are present in the surrounding environment [36,47]. In response to phytopathogenic bacteria, plant cells modulate different ionic signaling pathways [48] that eventually alter their plasma membrane potential [49,50,51]. Plant VOCs may induce membrane depolarization [52] and bacterial mVOC can travel far from the point of production through the atmosphere, porous soils, and liquid, making them ideal info-chemicals for mediating both short- and long-distance intercellular and organismal interactions [53]. 

#### 3.3.2. *E. amylovora* mVOCs Induce Calcium Efflux and Modulate K^+^ Channels of *A. thaliana*

In both plant–pathogen and plant–herbivore interactions, Vm depolarization has been associated with the Ca^+^-dependent opening of K^+^ channels [50,54]. Having assessed that the highest Vm depolarization occurs in early exposure time to *E. amlylovora* mVOCs, we measured by CLSM the cytosolic Ca^2+^ influx [Ca^2+^]_cyt_ with the fluorescent dye Calcium Orange^®^ and the activity of both voltage-gated K^+^ channels (VGKC) and ligand-gated or resting inward rectifier K^+^ channels (LG/RIRKC) with the fluorescent dye FluxOR^®^, with respect to control plants not exposed to *E. amlylovora* mVOCs.

To visualize the effect of *E. amylovora* mVOCs on [Ca^2+^]_cyt_, we calculated the ratio between the dye fluorescence of treatments and controls by expressing the fold change increase. Root caps showed the highest ratio after 30 min, whereas leaves and stems had the highest fold change after 1 h of mVOC exposure (Figure 3). The root elongation [Ca^2+^]_cyt_ was almost constant during the time-course experiment, whereas a decreasing trend was observed for leaves from 1 to 3 h (Figure 3). At 3 h of mVOC exposure, an increasing trend of [Ca^2+^]_cyt_ fold change was observed between roots and shoots (Figure 3).

Upon perception of bacterial and fungal elicitors, plant plasma membranes activate Ca^2+^ channels [50]. This event has been demonstrated for different kinds of elicitors, including oligopeptides, oligogalacturonides, beta-glucans, and symbiotic Nod factors [49,54]. Here we showed that *E. amylovora* mVOCs are able to induce a significant increase in [Ca^2+^]_cyt_ levels in both roots and shoots of the receiver *A. thaliana* seedlings. 

A different timing of K^+^ channel activity was found for VGKC between *A. thaliana* roots and shoots. The root cap and root elongation zone reached their highest fold change after 30 min, whereas leaves and stems showed the highest fold change after 2 h of exposure to *E. amylovora* mVOCs (Figure 4A). LG/RIRKC activity peaked after 1 h and 3 h in leaves, whereas in roots the activity remained constant up to 2 h and then declined at 3 h (Figure 4B). K^+^ efflux, which accompanies the observed Vm depolarization, is thought to occur via root and guard cell depolarization-activated K^+^ channels [55] and is known to react to microbial elicitors [56]. Elicitors from *E. amylovora* have been found to cause K^+^ efflux and cause Vm depolarization in *A. thaliana* [57]. Here we showed that also *E. amylovora* mVOCs are able to trigger the same cascade of events found in the plant–microbial [50,58], plant–plant [59] and plant–insect [60] interactions, with a cation influx (Ca^2+^ and K^+^) that causes Vm depolarization.

#### 3.3.3. *E. amylovora* mVOCs Induce a Strong Burst of ROS and NO in *A. thaliana* Seedlings

During the interaction of bacterial pathogens with plants, the generation of ROS and the oxidative burst play important roles in both disease progress and hypersensitive response (HR) development [61,62]. Because H_2_O_2_ and NO play a central role in disease resistance, we extended our analysis up to 16 h of exposure to *E. amylovora* mVOCs by evaluating the production of both ROS and NO. As for calcium and potassium channels, we expressed ROS and NO values as fold changes in the fluorescence of mVOC-exposed seedlings with respect to controls. The highest ROS fold change was found in root caps after 6 h exposure, whereas a decreasing trend was observed for both leaves and stems from 3 h to 16 h (Figure 5A). With regards to NO, leaves showed the highest fold change after 3 h (as for ROS), whereas the root caps showed an increasing trend of NO production, which peaked at 16 h (Figure 5B). Figure 6 summarizes the ROS and NO production in the roots and leaves of *A. thaliana* upon exposure to *E. amylovora* mVOCs. Stems and roots show the highest fluorescence between 3 and 6 h of mVOC exposure, with tissue localization in vascular tissues in the elongation zone and in the stems (Figure 6B) and in the root apex of secondary roots (Figure 6C). Almost the same localization was observed for NO both in roots (Figure 6E) and in shoots (Figure 6G).

In plant–pathogen interactions, two phases of ROS production are found: phase I is a nonspecifically stimulated transient ROS production by compatible (leading to disease), incompatible (leading to HR), and even saprophytic bacteria; whereas phase II is a delayed and prolonged ROS production that is specifically stimulated by incompatible HR-causing bacteria and is characteristic of the HR [63]. *E. amylovora* is known to induce both phases by inducing the production of ROS for several hours in both compatible and incompatible situations [64] and being able to multiply transiently in the non-host plant *A. thaliana,* where the production of intracellular ROS occurs during this interaction [65]. Our results show that *E. amylovora* mVOCs are able to induce both phases of ROS production, being able to trigger both early and late production of H_2_O_2_. Moreover, NO accumulated when *A. thaliana* was challenged by *E. amylovora* mVOCs, and NO is known to crosstalk with other signaling pathways by modulating HR and triggering defense responses [62] and is induced by microbial elicitors [66]. 

### 3.4. E. amylovora mVOCs Regulate the Gene Expression of A. thaliana Seedlings

Having assessed the significant modulation in plant growth, early signaling, and production of ROS and NO, we performed a full genome transcriptomic analysis of *A. thaliana* responses to *E. amylovora* mVOCs. Seven-day-old *A. thaliana* seedlings grown in HSCC were used for gene microarray analysis in the presence (for 16 h) or absence of *E. amylovora*. 

Analysis was performed in the full seedlings using an *A. thaliana* gene expression microarray 4 × 44 K (G2519, Agilent Technologies). Out of 43,663 analyzed genes, 8934 had a *p* < 0.05, and 155 of those showed a fold change difference relative to the control ≥2 or ≤0.5 (see Appendix A) and were used for further analyses. The list of DEGs satisfying the selection criteria along with the gene ontology functional categorization is reported in Appendix A).

We then analyzed the protein–protein association and interaction of the DEGs in Appendix A using STRING [30], considering known, predicted, and other types of interactions. A functional association was evidenced by k-means clustering that allowed to identify groups of proteins that contribute jointly to a specific biological function. Seven clusters were obtained by analyzing all the genes listed in Appendix A.

The first cluster is composed in general of defense genes and is made of a group of downregulated heat shock proteins and transcription factors and other upregulated proteins involved in pathogen response and detoxification (Figure 7I). Heat shock proteins and transcription factors are expressed in plants due to pathogenic interactions [67], and their expression has been reported to be induced through the accumulation of ROS [68]. Heat shock proteins 70 s have been shown to be involved in plant pathogen-associated molecular pattern-triggered immunity and effector-triggered immunity [69]. *BIP3* (*HSP70-13*) is classified as a marker gene of the *A. thaliana* unfolded protein response [70], and *BiP3* overexpression compromised immunity against *Xanthomonas oryzae* [71]. Small heat shock proteins such as ATHSP22.0, which was downregulated by *E. amylovora* mVOCs, are involved in priming defense against necrotrophic fungi [72].

The second cluster is characterized by genes that are mostly downregulated. These include a group of late embryogenesis abundant proteins (AT2G18340, AT3G02480, AT3G17520, LEA4-5, and LEA7) (Figure 7II). Late embryogenesis abundant (LEA) proteins accumulate to high levels during the late stage of seed maturation and in response to both abiotic and biotic stress, and they play a role in protecting plants from damage by protecting protein structure and binding metals under osmotic and oxidative stresses [73]. Some LEA proteins are involved in mitochondrial ROS signaling, impacting root development and pathogen responses [74]. The LEA5 group is also expressed in non-seed organs and was found to be induced by ABA treatment [75]. Downregulation of LEAs was associated with the downregulation of ABA-responsive genes, including two ABA-induced interacting protein phosphatases (*HAI2* and *HAI3*), a tryptophan-rich sensory protein-like protein (*TSPO*), a protein involved in ABA biosynthesis (*NCED3*), and a CAP160 protein (*LTI65*). HAI2 regulates seed dormancy by inhibiting ABA signaling [76] and has a functional role in ABA signaling and sugar tolerance in *A. thaliana* [77], whereas HAI3 plays a major role as a negative regulator of ABA signaling during early growth [78]. *AtTSPO* is targeted to the secretory pathway in plants, and expression of this gene may enhance ABA sensitivity [79]. TSPO was also associated with the induction of oxidative stress with a role in controlling redox homeostasis [80]. Finally, *NCED3* and *LTI65* were found to respond to isoprene, demonstrating that this VOC impacts ABA signaling at different tissue-specific, spatial, and temporal scales [81].

MKK7, a MAP kinase kinase, and RAB18 are also present in this cluster. *MKK7* has previously been shown to negatively regulate *A. thaliana* polar auxin transport and positively regulate plant basal and systemic acquired resistance (SAR) [82]; moreover, the increased expression of this gene was responsible for polar auxin transport deficiency leading to plant architectural abnormality [83]. RAB proteins belong to the small GTP-binding proteins. In *A. thaliana,* they are active in their GTP-bound form, which is membrane-associated following post-translational lipid modifications, i.e., prenylation [84]. *A. thaliana RAB18* is a stress-inducible gene that is significantly activated by ABA [85]. The only upregulated gene in this cluster is *RSL1*, which encodes a functional E3 ubiquitin ligase and whose overexpression alters hormonal responses in seed and vegetative tissues [86]. Overexpression of *RSL1* was found to reduce ABA sensitivity [87] (Figure 7, central left light green cluster). 

The above-described cluster is linked to a small cluster including a downregulated late embryogenesis abundant gene (*AT2G42560*) and two upregulated genes (an oleosin-like protein, *AT2G25890*, and a negative regulator of starch metabolic process, *QQS*) (Figure 7III). 

A strong downregulation was found for several seed storage albumins (*SESA1*, *SESA2*, *SESA3*, and *SESA5*) (Figure 7IV) that were clustered along with cruciferins (*CRU2* and *CRU3*). *A. thaliana* contains two predominant classes of seed storage proteins: napin-type albumins referred to as 2S albumin or arabin [88], including SESA1, SESA2, SESA3, and SESA5, and legumin-type globulins referred to as 12S globulin or cruciferin [89], including CRU2 and CRU3. In general, ABA plays a critical role in the regulation of seed maturation [90] and seed storage genes [91]. *SESA5*, the most downregulated gene in our analysis, is upregulated by ABA [92]. A cupin family protein (PAP85), which encodes a vicilin-like seed storage protein, is under the control of ABA [93] and was also downregulated, as were a hydroxysteroid dehydrogenase that responds to ABA (*HSD1*) [94] and a glycine-rich protein (*AT5G35660*) and a plant thionin (*AT2G15010*), which are involved in plant defense [95]. The only upregulated gene in this cluster codes for an uncharacterized protein (*AT1G27461*) that functions in DNA binding and is involved in seed dormancy [96]. 

Another cluster (Figure 7V) gathers proteins that are coded by both up and downregulated genes. Downregulation occurred for genes coding for terpene synthases: *AT3G25810*, that catalyzes the formation of eight monoterpenes from geranylpyrophosphate [97]; and *AT3G20160*, which encodes an active geranylgeranyl pyrophosphate synthase and is expressed in roots [98]. Downregulation occurred also for *AT3G58390*, a eukaryotic release factor protein that suppresses non-stop decay (NSD) and no-go decay (NGD) translation-coupled RNA quality control systems [99], and for *AT5G04120*, a phosphoserine phosphatase that forms serine from L-*O*-phosphoserine [100]. Upregulated genes coded for 1-deoxy-D-xylulose 5-phosphate synthase (DXPS1), a protein that either does not have DXS activity or whose activity is not enough to support full-rate isoprenoid biosynthesis [101]. Upregulation was also found for proteins involved in galactose oxidase (*AT1G60570*), pyruvate kinase (*AT4G26390*), alpha/beta-hydrolases (*AT2G19550*), a cysteine-rich secretory protein (*AT1G50060*), a ribosomal protein (*AT5G63070*), and a ubiquitin (*UBQ9*). 

This latter cluster was linked to a small cluster (Figure 7VI) characterized by transcripts of downregulated genes including *BSMT1*, which is involved in methyl salicylate biosynthesis [102] and is necessary for SAR signaling [103]; and *ELI3-2,* an aromatic alcohol:NADP^+^ oxidoreductase (benzyl alcohol dehydrogenase) involved in response to stress and lignin biosynthesis [104,105]; and two upregulated genes: *JAO4*, a jasmonic acid-coregulated 2-oxoglutarate dependent oxidase that is involved in response to biotic stress [106]; and *JOX3,* which has the ability to lower the jasmonate metabolism in vivo [107]. 

Finally, the last cluster (Figure 7VII) was composed of upregulated genes coding for nucleotide-binding protein (AT1G12805) and EXPA2, the latter is an α-expansin that plays a key role in controlling seed germination through gibberellic acid signaling [108]. Upregulation was also found for *GOX2*, which was found to be highly induced by programmed cell death and plays a role in the interplay between soluble sugars and ROS [109]. Downregulation occurred for *AT4G33905*, which codes for a stress- and ABA-regulated peroxisomal membrane protein [110]; and for *GASA3*, which codes for a GAST1 protein homolog 3 that responds to gibberellins [111]. The latter is linked through LEA7 and PAP85 to two other clusters. Finally, a type 4 metallothionein (*AtMT4a*) that is involved in regulating Zn ion accumulation in the late embryo and in controlling early seedling growth [112] is linked through PAP85 to the seed storage cluster (green cluster).

With regards to the other DEGs of Appendix A that were not clustered in the String analysis, a strong upregulation was found for *DAU2*, which is involved in gamete interaction that leads to correct double fertilization [113], and several genes expressed in the extracellular region involved in: response to oxidative stress and regulated by JA (*THI2.1*) [114], defense response to the bacterium (*LCR6*) and to ABA (*AT5G24080*). Among the downregulated DEGs, some were involved in response to ABA and a few in defense response. With regards to DEGs responding to ABA, *AT5G05220* is also a hydrogen peroxide-inducible protein [115], whereas *AT5G45630* responds to ABA and salicylic acid and is modulated by bacterial infection [116]. Among defense genes, *DIP2* is involved in viral resistance [117], whereas *AT4G03480* is an ankyrin repeat-containing protein that mediates protein–protein interactions [118]. Validation of microarray data was done by selecting three of the most downregulated genes (*SESA1*, *SESA2,* and *SESA3*), which were evaluated in a time-course experiment. A hormetic trend was found for all genes, with peaks of expression at 9 and 12 h (Appendix A).

### 3.5. E. amylovora mVOCs Modulate Auxin Efflux Carriers PIN1 and PIN3 of A. thaliana Roots

*E. amylovora* mVOCs were found to enhance plant productivity and modulate root morphology of exposed *A. thaliana* seedlings. Therefore, because auxin plays a key regulatory role in plant growth and root morphology [119], we assayed the gene expression and localization of components of the auxin efflux machinery mediating polar auxin transport, PIN1 and PIN3 [120]. PIN1 mainly resides at the basal end of the vascular cells and regulates root hair initiation areas in the root epidermal cells [121] (see Figure 8C), while PIN3 is localized without polarity in the columella cells, predominantly basally in vascular cells, and laterally in pericycle cells of the elongation zone [122] (see Figure 8E). GUS staining of transgenic plants carrying the PIN1::GUS construct in the root meristem region of plants exposed to *E. amylovora* mVOCs for 16 h (Figure 8B) shows a higher staining in the vascular cells above the quiescence center with respect to control unexposed roots (Figure 8A). In PIN3::GUS mutants, we found a stronger expression in the lateral root cap in mVOCs-treated plants (Figure 8E) with respect to controls (Figure 8D).

We then evaluated the gene expression of *PIN1* (*AT1G73590*) and *PIN3* (*AT1G70940*) in the roots and shoots of control and mVOC-exposed *A. thaliana* seedlings in time-course experiments. Although the upregulation of *PIN1* and *PIN3* was not above two-fold with respect to controls, at early times of exposure root and leaf *PIN1* and root *PIN3* were significantly downregulated. The expression of root *PIN1* increased with time of exposure to mVOCs with respect to controls (Figure 9), whereas in leaves the highest fold change was found after 9 h. The same trend was observed for root *PIN3* gene expression, but leaves did not show significant upregulation (Figure 9).

Intercellular transport of auxin is essential for root morphogenesis, and here we showed that expression of *A. thaliana* PIN1 and PIN3 auxin transporters is enhanced by exposure to *E. amylovora* mVOCs. Biotic stress has been shown to alter PIN expression [123], and root-interacting beneficial and pathogenic microorganisms utilize auxin and its target genes to manipulate the performance of their hosts for their own needs [124,125]. Increased auxin levels caused by plant–bacterial interactions have been shown to alter root morphology and increase plant resistance to abiotic stress [126]. Although the upregulation of *PIN1* and *PIN2* was not so evident (fold-change below 2), a clear increasing trend in their gene expression was observed with time upon exposure to mVOCs.

Very recently, a link between ROS and both auxin distribution and the response has been observed in *A. thaliana* roots, with ROS production being necessary for orchestrating cell division and auxin flux during root development [127]. Therefore, our results confirm that ROS signaling is involved in auxin efflux during *A. thaliana’s* perception of *E. amylovora* mVOCs.

### 3.6. E. amylovora Synthetic mVOCs 1-nonanol and 1-dodecanol Alter Morphology and Induce Oxidative Stress in A. thaliana Seedlings

Although we are aware that evaluating identified mVOCs individually using their synthetic versions may miss VOC combinations that work synergistically [19], we tested two molecules identified in *E. amylovora* mVOCs and listed in Table 1: 1-nonanol and 1-dodecanol, and evaluated their role in plant morphology and ROS gene expression.

Eight-day-old *A. thaliana* seedlings were exposed to either 1-nonanol or 1-dodecanol at 20 µg, 50 µg, and 100 µg for 72 h, and the root length was measured before and after exposure to the synthetic mVOCs. These concentrations were considered by referring to literature data [13,14]. DMSO was used as a solvent. No significant difference was found among controls treated with DMSO. After 72 h exposure to either 1-nonanol or 1-dodecanol, we observed an induction of root growth (*p* < 0.05) when the synthetic molecules were used at 20 µg and 50 µg, whereas no significant differences were found for either of them at 100 µg (Figure 10). 

As reported above, 1-nonanol was found to be associated with *A. thaliana* lateral root formation, suggesting a participation via jasmonic acid (JA) signaling [31], and our results confirm its promoting action on *A. thaliana* roots. We also measured the density of secondary root production, and we noticed a significant increase in secondary root emission (see Appendix A).

Because we showed that auxin is involved in root mVOC response and considering that ROS are involved in cell division and auxin flux during primary and secondary root development [125], we assessed the expression of some key genes coding for enzymes involved in ROS production. We quantified the expression of the respiratory burst oxidase homolog *RBOHH*, which codes for a plasma membrane-located NADPH oxidase that produces the superoxide anion (O_2_^•−^), and of Cu/Zn-superoxide dismutase 1 (*SOD1*), which codes for a protein that reduces O_2_^•−^ to H_2_O_2_ [128]. The latter molecule is detoxified to water by a number of scavenging enzymes, including catalase (*CAT1*), ascorbate peroxidases (*APX1*), and anionic peroxidase (*PER4*) [129,130].

In roots exposed to 1-nonanol (Figure 11A), *RBOHH* was downregulated at all 1-nonanol concentrations, whereas the other genes did not show a regulation higher than 2 folds, with the exception of *SOD1,* which was downregulated by exposing roots to 50 µg 1-nonanol. Shoots exposed to both 20 µg and 100 µg 1-nonanol always showed a strong gene downregulation, particularly for *RBOHH* (Figure 11B). Interestingly, exposure to 50 µg 1-nonanol strongly upregulated the tested genes, with *CAT1* being the most upregulated (Figure 11B). Exposure to 1-dodecanol always downregulated *RBOHH* and showed for the other genes an upregulation at 20 µg and 100 µg and the same response as controls at 50 µg (Figure 11B). Finally, shoots exposed to 1-dodecanol showed a strong downregulation for *RBOHH* at all concentrations and no significant variations with respect to controls for almost all genes, with the exception of *PER4,* which was upregulated by 50 µg and 100 µg 1-dodecanol. 

A direct comparison of the effects of the two synthetic mVOCs indicates a common downregulation of *RBOHH* at all tested concentrations. This gene is regulated by cytosolic calcium and protein phosphorylation, and its H_2_O_2_ production was found to dampen growth rate oscillations [131]; furthermore, some RBOHs have been found to activate ROS signaling [132]. Another common response is the lower regulation in roots with respect to shoots (see the different scales of Figure 11A,C vs. Figure 11B,D), with shoots appearing more sensitive to the presence of the synthetic mVOCs. A stronger root response was found for *SOD1* in roots exposed to 1-dodecanol (Figure 12C) with respect to 1-nonanol (Figure 12A). SOD1 is cytosolic [133], and the overexpression of *SOD1* increases the tolerance to pathogens in many plant species, including *A. thaliana* [134,135]. Shoots of seedlings exposed to both synthetic mVOCs show a strong upregulation of *PER4*, and the same occurred in roots exposed to 1-dodecanol. NO production activates the transcription of the defense-related gene *PER4,* which is putatively located in the endomembrane system and involved in the mechanisms underlying *A. thaliana* basal resistance to the phytopathogen *Botrytis cinerea* [136]. *CAT1* was strongly upregulated in shoots exposed to 1-nonanol, and in *A. thaliana* mutants with increased levels of ROS, CAT1 activity was found to partially rescue growth [137]. Although APX1, an important defense enzyme, protects plant cells from disease agents via systemic acquired resistance [138], its regulation was almost always similar in controls and plants exposed to the synthetic mVOCs. 

## 4. Conclusions

By using the model system HSCC, we showed that the phytopathogen *E. amylovora* emits mVOCs that are able to enhance *A. thaliana* growth and trigger cascades of both early and late events. The complex blend of mVOCs included volatile hydrocarbons, alkenes, alkanols, and alkanals, including 1-nonanol and 1-dodecanol that we tested as synthetic molecules. The results of this work show that the interactions between *E. amylovora* mVOCs and *A. thaliana* result in morphological and molecular changes. Perception of mVOC by *A. thaliana* was found both in roots and shoots, with the activation of early signaling events including cation channels (Ca^2+^ and K^+^) activity and plasma membrane (Vm) depolarization. These signaling pathways triggered a strong oxidative burst that was caused by the increased levels of both H_2_O_2_ and NO. Transcriptomic analyses allowed to identify hundreds of DEGs that were further processed by protein–protein association and interaction revealing the presence of genes responding to phytohormones, with particular reference to ABA, gibberellic acid, and auxins, the latter being associated with the modulated root morphology and growth. The increased plant growth was related to the strong downregulation of seed storage proteins and late embryogenic abundant proteins, suggesting that the bacterium mVOCs might play a role in the mobilization of plant reserves. We hypothesize that this could be a strategy by the microorganism to obtain a higher availability of resources. Finally, we tested two among the various mVOCs produced by *E. amylovora* (1-nonanol and 1-dodecanol), and we confirmed that they are effective in root promotion and in ROS production and scavenging. The differential modulation by the two molecules confirms the hypothesis that mVOCs act in a synergistic way and further studies are needed to assess the specific role of each individual mVOC composing the microbially emitted blend. However, it has to be considered that not all mVOC molecules, which have bioactive properties, may influence *A. thaliana* growth through the same mechanism and that a specific receptor might exist. In fact, some mVOCs could be transported into the cytoplasm through the plasma membrane, especially if they are present in the extracellular space in gaseous form. 

The scheme of Figure 12 summarizes the early and late events occurring in *A. thaliana* seedlings upon perception of *E. amylovora* mVOCs.

## Figures and Tables

**Figure 1 antioxidants-12-00600-f001:**
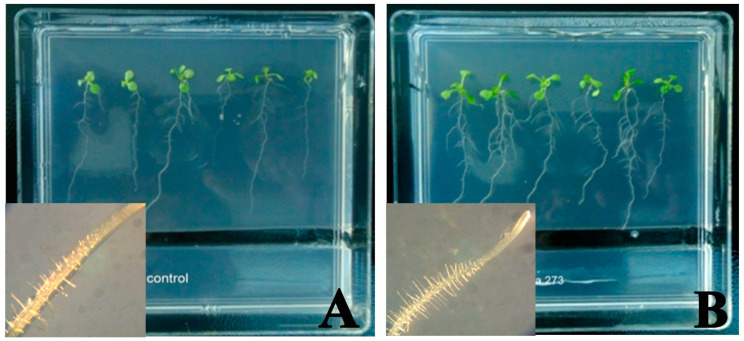
*Arabidopsis thaliana* seedlings responses to *Erwinia amylovora* mVOCs. (**A**) Control Petri capsule with only *A. thaliana* seedlings and the two culture media (the top one for plant and the bottom one for bacterial growth). The insect shows a typical root in control plants. (**B**) *A. thaliana* seedlings of the same age as (**A**) but exposed for 8 days to *E. amylovora* mVOCs. The inset shows a typical *A. thaliana* root exposed to mVOCs. The width of the Petri dish is 100 mm.

**Figure 2 antioxidants-12-00600-f002:**
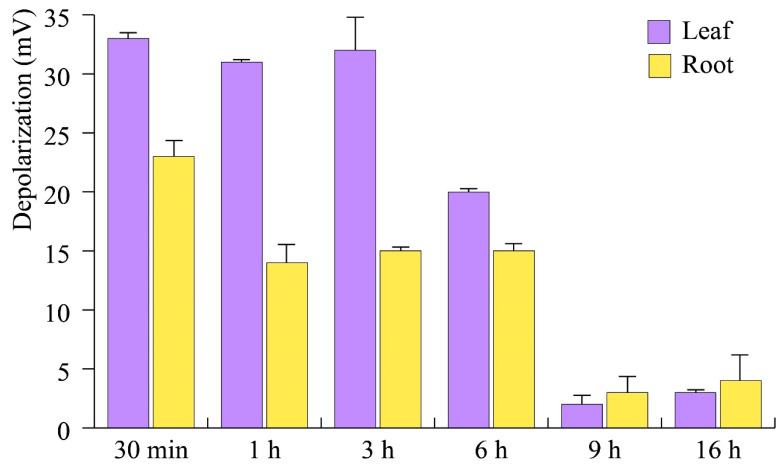
Plasma membrane potential (Vm) depolarization time course analysis in roots and leaves of *A. thaliana* seedlings exposed to *E. amylovora* mVOCs. Metric bars indicate standard deviation.

**Figure 3 antioxidants-12-00600-f003:**
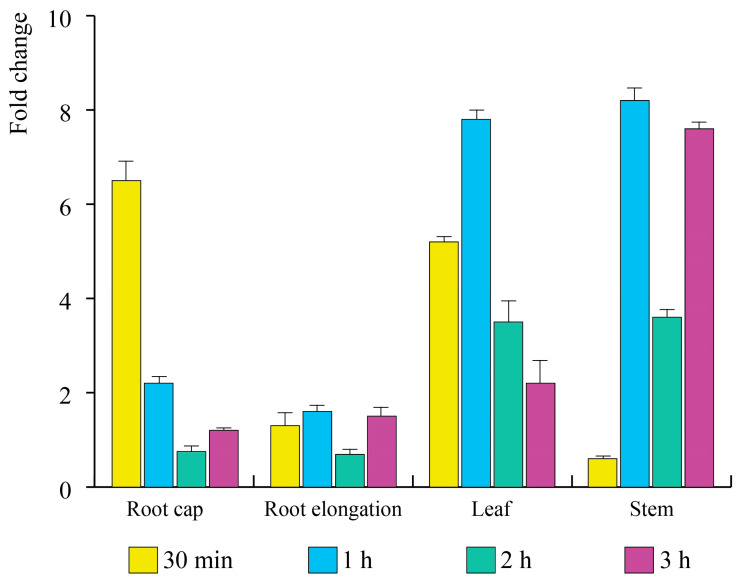
Time-course changes in [Ca^2+^]_cyt_ in roots and shoots of *A. thaliana* seedlings exposed to *E. amylovora* mVOCs. Values are expressed as the fold change between the fluorescence of treatments vs. controls (see Materials and Methods). Bars represent standard deviation.

**Figure 4 antioxidants-12-00600-f004:**
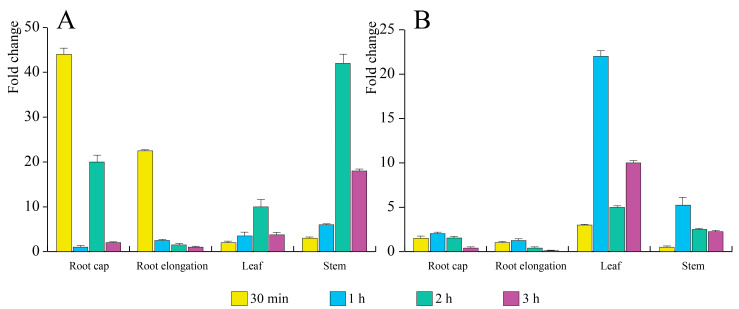
Time-course changes in (**A**) Voltage gated K^+^ channels (VGKC) and (**B**) Ligand gated/resting inward rectified channels (KG/RIRKC) in roots and shoots of *A. thaliana* seedlings exposed to *E. amylovora* mVOCs. Values are expressed as the fold change between the fluorescence of treatments vs. controls (see Materials and Methods). Bars represent standard deviation.

**Figure 5 antioxidants-12-00600-f005:**
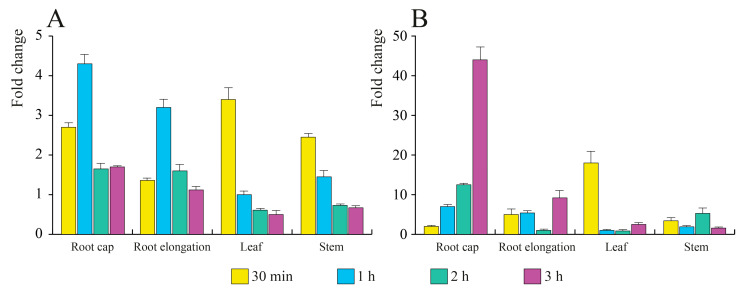
Time-course changes in ROS (**A**) and NO (**B**) in roots and shoots of *A. thaliana* seedlings exposed to *E. amylovora* mVOCs. Values are expressed as the fold change between the fluorescence of treatments vs. controls. Bars represent standard deviation.

**Figure 6 antioxidants-12-00600-f006:**
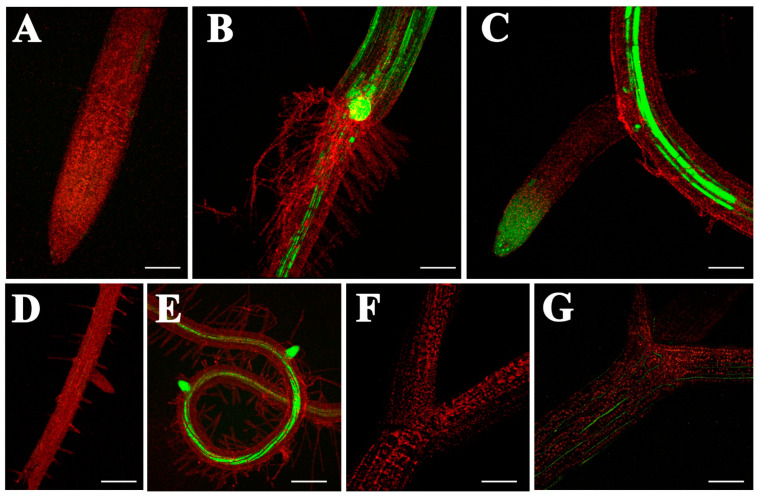
CLSM images of ROS (**A**–**C**) and NO (**D**–**G**) localization in *A. thaliana* seedlings upon exposure to *E. amylovora* mVOCs. (**A**) control root of a plant grown without the presence of mVOCs and stained with Amplex Red. (**B**) Yellow fluorescence of ROS and active peroxidases in a plant exposed for 6 h to mVOCs showing a portion of the junction between the root and the shoot, where the yellow fluorescence refers to the reaction of 10-acetyl-3,7-dihydroxyphenoxazine (Amplex Red) with H_2_O_2_, whereas the root and shoot tissues not reacting with Amplex Red are shown in bright red color. (**C**) Portion of a root exposed for 6 h to mVOCs showing the presence of a secondary root with a clear reaction with the dye Amplex Red both in the secondary root tip and in the vasculature of the main root. (**D**) Portion of a control root perfused with carboxy-2-phenyl-4,4,5,5-tetra-methylimidazolinone-3-oxide-1-oxyl (cPTIO), an NO scavenger, before being stained with 4,5-diaminofluorescein diacetate (DAF-FM DA) in plants exposed for 16 h to mVOCs. (**E**) Reaction of NO with DAF-FM DA in roots of plants exposed for 16 h to *E. amylovora* mVOCs; a bright yellow fluorescence indicates the reaction in the root vascular tissues and in the secondary root tips. (**F**) Control stems treated as in (**E**) but after 3 h exposure to mVOCs. (**G**) Vascular tissues of *A. thaliana* stems show an NO localization after 3 h of exposure to *E. amylovora* mVOCs. Metric bars: 250 µm.

**Figure 7 antioxidants-12-00600-f007:**
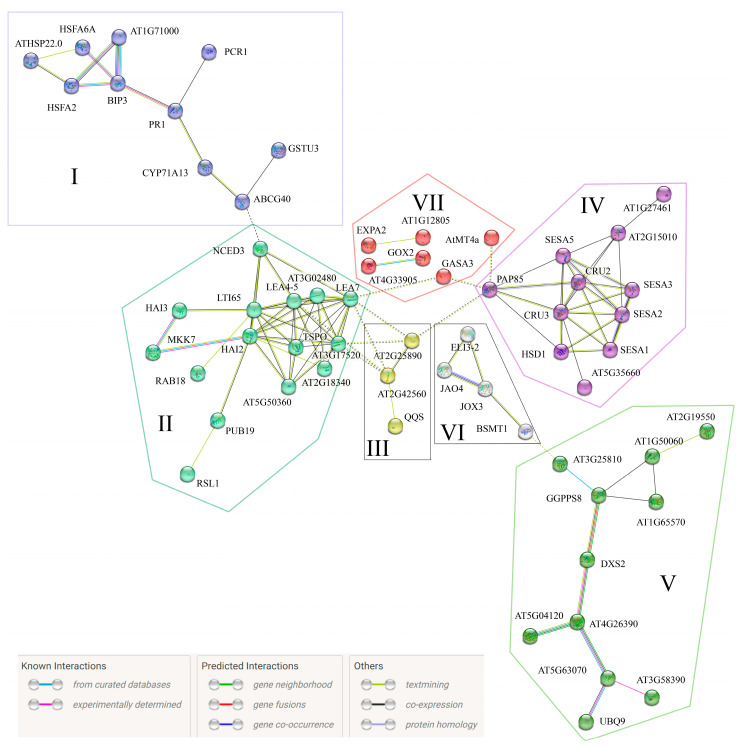
Association networks between proteins coded by the DEGs of Appendix A obtained with STRING [30]. Nodes indicate the different proteins. Associations are meant to be specific and meaningful, i.e., proteins jointly contribute to a shared function; this does not necessarily mean they are physically binding to each other. Color codes of known, predicted and other interactions are indicated at the bottom of the figure. Different node colors indicate proteins clustered by k-means clustering. Edges between clusters are indicated by dotted lines, and the confidence cutoff for showing interaction links has been set to “highest” (0.900). Roman numbers (**I**–**VII**) refer to the different clusters.

**Figure 8 antioxidants-12-00600-f008:**
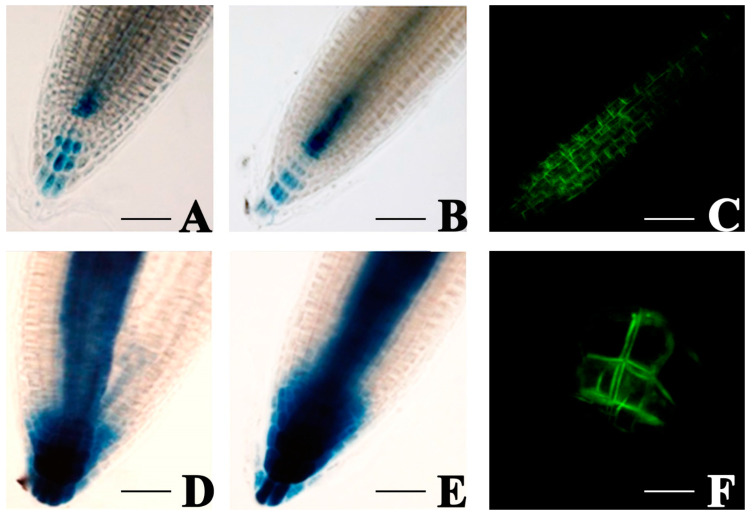
Localization of PIN1 (**A**–**C**) and PIN3 (**D**–**E**) efflux carriers in *A. thaliana* roots. (**A**) GUS staining of transgenic plants carrying PIN1::GUS construct efflux carriers in control roots; (**B**) GUS staining of transgenic plants carrying PIN1::GUS construct efflux carriers in roots exposed to *E. amylovora* mVOCs for 9 h; (**C**) CLSM of PIN1:GFP efflux carriers in roots exposed for 16 h to *E. amylovora* mVOCs; (**D**) GUS staining of transgenic plants carrying PIN3::GUS construct efflux carriers in control roots; (**E**) GUS staining of transgenic plants carrying PIN3::GUS construct efflux carriers in roots exposed to *E. amylovora* mVOCs for 9 h; (**F**) CLSM of columella localized PIN3:GFP efflux carriers in roots exposed for 9 h to *E. amylovora* mVOCs. (See also Appendix A for DR5::GUS, PIN1:GFP, and PIN3:GFP controls). Metric bars: (**A**–**E**), 200 µm; (**F**) 100 µm.

**Figure 9 antioxidants-12-00600-f009:**
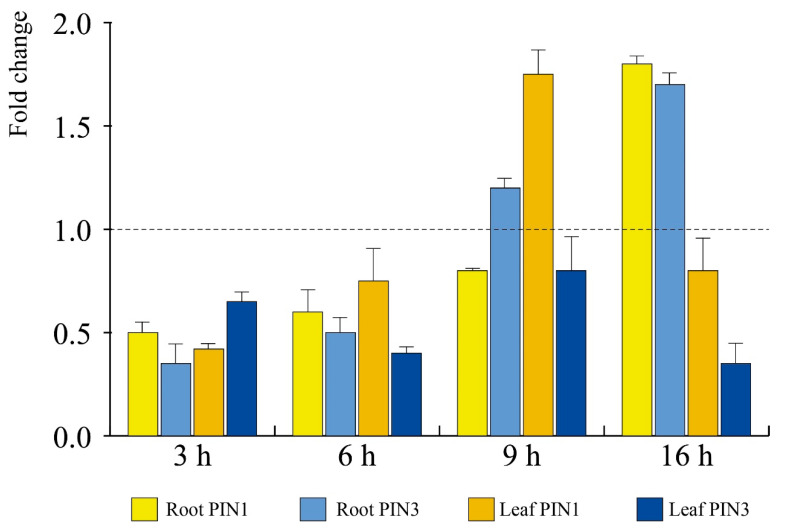
Expression of *PIN1* (*AT1G73590*) and *PIN3* (*AT1G70940*) genes in time-course exposure of *A. thaliana* shoots and roots to *E. amylovora* mVOCs. Data are expressed as fold change with respect to the controls (DMSO). Values above or below the dotted line indicate upregulation and downregulation, respectively. Bars indicate standard deviation.

**Figure 10 antioxidants-12-00600-f010:**
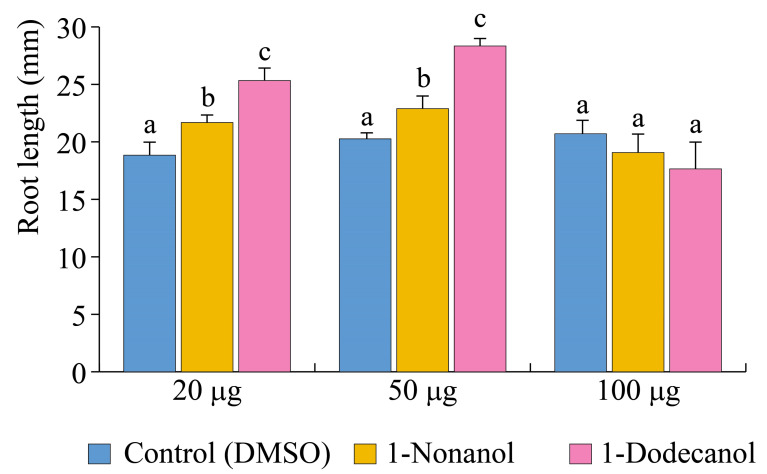
Effect of exposure to synthetic mVOCs 1-dodecanol and 1-nonanol on root length of *A. thaliana* seedlings. Metric bars indicate standard deviation, different letters indicate significant (*p* < 0.05) differences.

**Figure 11 antioxidants-12-00600-f011:**
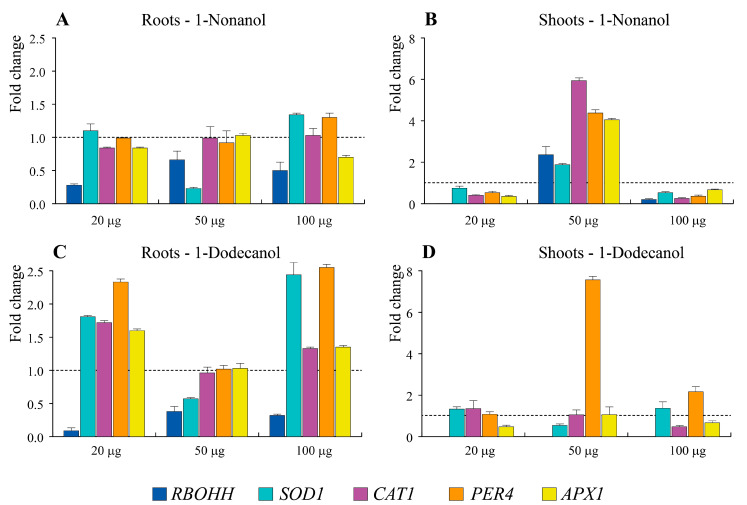
Gene expression of *RBOHH*, *SOD1*, *CAT1*, *PER4,* and *APX1* in *A. thaliana* roots (**A**,**C**) and shoots (**B**,**D**) exposed for 72 h to 1.nonanol (**A**,**B**) and 1-dodecanol (**C**,**D**). Data are expressed as fold change with respect to control (DMSO, dotted line). Values above or below the dotted line indicate upregulation and downregulation, respectively. Bars indicate standard deviation.

**Figure 12 antioxidants-12-00600-f012:**
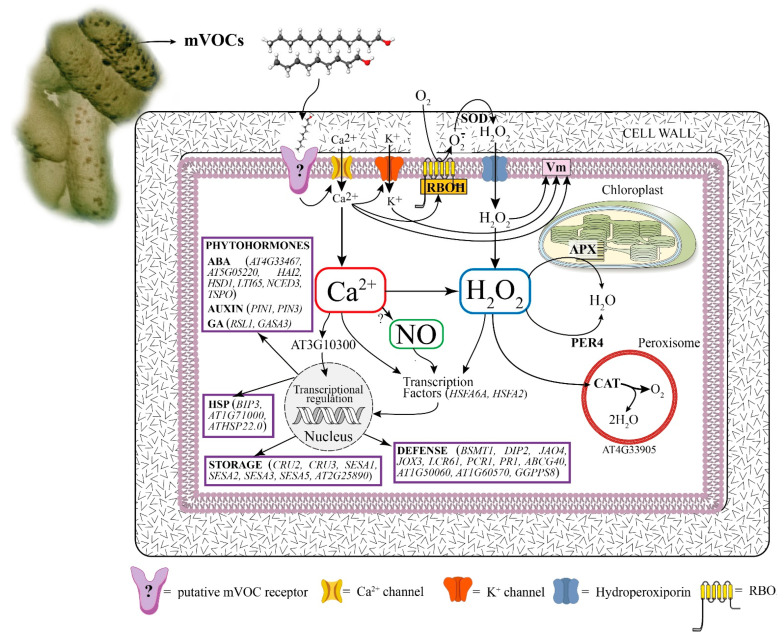
Early and late events in *A. thaliana* cells upon exposure to *E. amylovora* mVOCs. mVOCs are perceived at the plasma membrane by a still uncharacterized receptor/sensor, which eventually initiates a cascade of events involving the opening of Ca^2+^ channels. The increased Ca^2+^ cytosolic concentration triggers the opening of ligand-gated or resting inward rectifier K^+^ channels that, along with Ca^2+^, depolarize the membrane potential (Vm). These events are followed by the production of H_2_O_2_ by RBOHH and SOD, which increases the cell and tissue content of ROS. The latter are scavenged by CAT, APX, and PER4 that generate H_2_O and O_2_ from H_2_O_2_. The increased cellular ROS (and probably the increased cytosolic Ca^2+^) increase the NO content. All of these early events induce expression of genes involved in defense and storage, and those responding to phytohormones and heat stress.

**Table 1 antioxidants-12-00600-t001:** Chemical composition of *E. amylovora* mVOCs. Values are expressed as relative percentage and ordered according to the Kovàts retention index. See also Appendix A for GC mass spectra data.

Compound	Kovàts Retention Index	Relative Percentage
Iso amyl alcohol	697	1.21
Styrene	883	4.04
6-Methyl-2-heptanone	888	0.93
4-Octen-3-one	960	2.91
Benzaldehyde	982	1.13
Octanal	1005	1.08
Tetraisobutylene	1030	1.93
(E)-2-Octenal	1036	1.96
1,8-Cineole	1058	0.31
1-octanol	1065	2.58
Nonanal	1104	1.32
1-Nonanol	1159	3.03
Dichloromethyl propyl sulfone	1172	5.54
*trans*-2-Decenal	1212	2.16
Methylcyclodecane	1260	3.33
*trans*-2-Decenol	1266	2.36
Undecanal	1303	2.32
Tridecane	1313	1.08
6,10-Dimethyl-2-undecanone	1321	8.32
1-Undecanol	1357	2.08
2-Ethyl-1-decanol	1393	0.39
1-Tetradecene	1403	0.94
E-2-dodecenal	1410	0.48
Tetradecane	1413	1.21
1-Dodecanol	1457	10.10
Tridecanal	1502	2.21
Pentadecane	1512	11.73
Tetradecanal	1601	1.38
Tetradecanal	1601	2.12
2-Pentadecanone	1648	1.23
1-Tetradecanol	1656	1.94
Pentadecanal	1701	6.19
Heptadecane	1711	2.68
6,10,14-Trimethyl-2-pentadecanone	1754	6.48

## Data Availability

Data are provided in Appendix A and are available on request.

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
