# Peer review of "Bacterial Volatiles (mVOC) Emitted by the Phytopathogen Erwinia amylovora Promote Arabidopsis thaliana Growth and Oxidative Stress"

_antioxidants, 2023, doi:10.3390/antiox12030600_

Round 1

Reviewer 1 Report

The manuscript presents very interesting and novel data that the volatile organic compounds of Erwinia amylovora enhance Arabidopsis thaliana shoot and root. The analytical and statistical analysis and the graphical presentation are clear. The manuscript is well written.

The volatile organic compounds of Erwinia amylovora improve the growth of the shoot and roots of Arabidopsis thaliana. In my opinion, the topic is original in the field of plant physiology. The results present novelty because there is still an unclear microbial volatile organic compound affect on plant growth through a signaling event that involve phytohormones (i.e. auxins, abscisic acids, and gibberellins) and reactive oxygen species. In my opinion, the methodology was presented clearly. The conclusions are consistent with the evidence. The references are appropriate. The table and figures are clearly presented.

Author Response

we thank the reviewer for appreciating our work

Reviewer 2 Report

In this manuscript the authors present several results on the effects of the phytopathogen E. amylovora on the physiology of Arabidopsis. Several molecular, biochemical and microscopic methods were applied and led to large amounts of data and results. Nevertheless, in particular cases the way of presentation and also of interpretation needs to be revised:

- In respect to the title of the paper I would expect some quantitative data about the growth promoting effects of Erwinia mVOCs. Except Fig. 1 nothing is shown about it. Two graphs supporting this figure with quantitative data about root (analogous to Fig. 10) and shoot growth would be essential in this regard.

- Please rearrange the bar charts in Figs. 3-5 according the tissue and not to the time. Then, the tendencies should be seen clearer.

- Split Fig. 6 into two Figs. (A-C and D-G).

- Mark the different clusters in Fig. 7 with lines around them and give them roman numbers, which should also appear in the main text.

- Please shift Table 2 to the supplements. It destroys the readability of the manuscript.

- The complete section 3.5 is highly controversial. Fig. 8 A, B, D, and E should show that mVOCs lead to higher PIN gene expression (by GUS staining). Sorry, but I cannot see this. Additionally, Figs. 8C and F don´t confirm anything in this regard. In Fig. 9 root expression (RT-PCR) seems to be higher after 16 h, but this tendency is still below a 2-fold change. Depending on the applied qRT system this would not meet the criteria for significant changes. Therefore, I would ask for more convincing data in this respect. Alternatively, you shouls mark the effects on PIN gene expression as contradictory.

- I´m missing the indication of p-values (e. g. as asterisks) in Fig. 10 as measures of significance.

- Due to the number of data Fig. 11 is difficult to interpret. Please mark over A/C and B/D as "Roots" and "Shoots", respectively. Furthermore, mark at right hand side either 1-nonanol or 1-dodecanol.

Author Response

We thank the reviewer for providing us the way to greatly improve our manuscript. We followed the reviewer’s suggestions as reported step-by-step below.

- In respect to the title of the paper I would expect some quantitative data about the growth promoting effects of Erwinia mVOCs. Except Fig. 1 nothing is shown about it. Two graphs supporting this figure with quantitative data about root (analogous to Fig. 10) and shoot growth would be essential in this regard.

R: we thank the reviewer for suggesting to provide more quantitative data. We calculated the root length, secondary root production and shoot increased area by image analysis and provided quantitative data and statistics in the text.

- Please rearrange the bar charts in Figs. 3-5 according the tissue and not to the time. Then, the tendencies should be seen clearer.

R: We thank the review for this important suggestion. Figures 3, 4 and 5 have been redrawn and now the tendencies are clearer

- Split Fig. 6 into two Figs. (A-C and D-G).

R: The aim of figure 6 is to represent in a common figure the localization of both ROS and NOS; therefore, we would like to keep it together in order to provide the reader a common view and a comparative picture, without referring to two different figures. Therefore, we would like to keep Figure 6 as it is.

- Mark the different clusters in Fig. 7 with lines around them and give them roman numbers, which should also appear in the main text.

R: We thank the reviewer for this suggestions that surely improved both the visibility and the discussion of the clustering. Figure 7 has been edited and added with squares and Latin numbering that refer to the clusters and the related numbering has been reported in the text.

- Please shift Table 2 to the supplements. It destroys the readability of the manuscript.

R: Although the Table 2 represents the backbone of our transcriptomic analysis we accepted the suggestion of the reviewer to move it to supplementary data. Therefore any reference to the Table 2 is now addressed to Supplementary Table S2.

- The complete section 3.5 is highly controversial. Fig. 8 A, B, D, and E should show that mVOCs lead to higher PIN gene expression (by GUS staining). Sorry, but I cannot see this.

R: Indeed plants exposed to mVOC show a higher GUS staining which is evident in both PIN1 and PIN 3 mutants. We added a sentence to the text to better explain this difference.

Additionally, Figs. 8C and F don´t confirm anything in this regard.

R: We thank the reviewer for raising this issue. We addressed the two figures to the description of the localization of PINs in the text.

In Fig. 9 root expression (RT-PCR) seems to be higher after 16 h, but this tendency is still below a 2-fold change. Depending on the applied qRT system this would not meet the criteria for significant changes. Therefore, I would ask for more convincing data in this respect. Alternatively, you should mark the effects on PIN gene expression as contradictory.

R: we thank the reviewer for raising this point. Actually, the intention of this part was to show the increasing trend of gene expression. Although we agree that the higher fold change is lower than 2 for upregulation, nevertheless at early times both PIN1 and PIN3 are dowregulated below 0.5 (i.e., more than 2-fold) and there is an increasing gene expression with time that confirms our hypothesis. We modified the text and added a sentence to better explain this point.

- I´m missing the indication of p-values (e. g. as asterisks) in Fig. 10 as measures of significance.

R: we thank the reviewer for noticing this missing information. The full statistics has been added to figure 10.

- Due to the number of data Fig. 11 is difficult to interpret. Please mark over A/C and B/D as "Roots" and "Shoots", respectively. Furthermore, mark at right hand side either 1-nonanol or 1-dodecanol.

R: we thank the reviewer for these remarks that helped us to improve the figure. Figure 11 now reports both the organs and molecules as suggested.

Reviewer 3 Report

The review of the paper written by Parmagnani et al., submitted into Antioxidants MDPI. The main aim of this paper is to verify the influence of mVOC emitted by Erwinia amylovora on growth and oxidative stress in Arabidopsis plant. The authors performed well described experiments, which verified stated hypothesis and led them to the conclusion about strong oxidative burst due to calcium and potassium ions flux and H2O2 and NO production in response to mVOC, which should be recognize by some specific receptor in plasma membrane, what was suggested in Figure 13. I have positive feedback regarding this paper, however a few questions/doubts erased in my mind:

-          As the plants were grown together with the bacteria on bipartite squared Petri dish I have been wondering about the concentration of mVOC. Was it check in some way to be sure, that the concentration was the same or standardized in other way (the same number of plants, the same amount of bacteria…).

-          The Figure 13 suggests that all mVOC molecules, which have bioactive properties, influence Arabidopsis growth through the same mechanism, by Ca2+, K+, H2O2, NO. I am not sure about it. I also doubt if there was a specific receptor for each mVOC molecules. Some of them could be transported into cytoplasm through plasma membrane, especially if they are present in extracellular space as gas form.

I have also some remarks to the manuscript:

-          In lines 285-287 the authors stated that the presence of 1,8-cineole in bacteria is described for the first time in literature. I advise the authors to check the paper by Dr. Chiaki Nakano, Hyo-Kyoung Kim, Prof. Yasuo Ohnishi published in 2011 doi: 10.1002/cbic.201100330, Identification of the First Bacterial Monoterpene Cyclase, a 1,8-Cineole Synthase, that Catalyzes the Direct Conversion of Geranyl Diphosphate.

-          The minor once are:

-          L67 Volatile Organic Compounds should be with lower case

-          L81, L238, L303, L304, L305, L417 Arabidopsis should be with italics or lower case

-          L118 Calcium Orange lower case

-          L348 K+ upper index for +

-          I ask the authors to try increase the quality of Figure 7

Author Response

We thank the reviewer for the appreciation of our work and for providing important comments that helped us to improve the manuscript.

-          As the plants were grown together with the bacteria on bipartite squared Petri dish I have been wondering about the concentration of mVOC. Was it check in some way to be sure, that the concentration was the same or standardized in other way (the same number of plants, the same amount of bacteria…).

R: we thank the reviewer for raising this important point. A new sentence in the materials and methods clarifies that “In order to standardize the experiments, the same number of seeds and the same amount of E. amylovora CFU were always used”

-          The Figure 13 suggests that all mVOC molecules, which have bioactive properties, influence Arabidopsis growth through the same mechanism, by Ca2+, K+, H2O2, NO. I am not sure about it. I also doubt if there was a specific receptor for each mVOC molecules. Some of them could be transported into cytoplasm through plasma membrane, especially if they are present in extracellular space as gas form.

R: we agree with the remarks of the reviewer about the generalization of the Scheme of Figure 13 (now corrected to Figure 12). We added a sentences that takes in consideration the fact that not all mVOCs might act in the same way.

I have also some remarks to the manuscript:

-          In lines 285-287 the authors stated that the presence of 1,8-cineole in bacteria is described for the first time in literature. I advise the authors to check the paper by Dr. Chiaki Nakano, Hyo-Kyoung Kim, Prof. Yasuo Ohnishi published in 2011 doi: 10.1002/cbic.201100330, Identification of the First Bacterial Monoterpene Cyclase, a 1,8-Cineole Synthase, that Catalyzes the Direct Conversion of Geranyl Diphosphate.

R: we thank very much the reviewer for suggesting this important references that we overlooked. A new sentence now reports the presence of the sesquiterpene cyclase in S. clavuligerus that is able to convert GGPP into 1,8-cineole.

-          The minor once are:

-          L67 Volatile Organic Compounds should be with lower case

R: done

-          L81, L238, L303, L304, L305, L417 Arabidopsis should be with italics or lower case

R: we changed the term Arabidopsis with the latin name A. thaliana throughout the text

-          L118 Calcium Orange lower case

R: done

-          L348 K+ upper index for +

R: done

-          I ask the authors to try increase the quality of Figure 7

R: we thank the reviewer for this remark. We redraw figure 7 also according to the reviewer 2 suggestions and we increased the font size in order to obtain a better visibility of protein labels.

Round 2

Reviewer 2 Report

Now I am fine with it, thanks.

Reviewer 3 Report

The authors answered all my questions and modified the manuscript accordingly.